# The comparison of comprehensive sexuality education knowledge and attitudes in CSE-exposed and CSE-naïve non-formal settings in Lagos State, Nigeria

**Babatunde ADELEKAN**[1], **Esther SOMEFUN**[1], **Olushola KAREEM**[1], **Oladimeji IBRAHIM**[1], **Hamira WELYE**[1], **Rabiatu SAGEER**[1], **Erika GOLDSON**[1], **Rashidat UMAR**[2], **Adekemi Oluwayemisi SEKONI**[3]*

1 United Nations Population Fund, Nigeria Country Office, Abuja, Nigeria, 2 Lagos State Ministry of Youth and Social Development, Lagos, Nigeria, 3 College of Medicine Lagos, University of Lagos, Lagos, Nigeria

* asekoni@unilag.edu.ng

## Abstract

This study compared the sexual reproductive health and rights related knowledge and attitudes towards inclusive gender norms among students in comprehensive sexuality education (CSE) exposed and naïve technical and vocational centers in Lagos state, Nigeria. It also explored the benefits of teaching and learning about CSE from exposed students and their instructors. A mixed-method research design was used. The study population comprised of students 15–24 years of age attending technical and vocational centers in Lagos State and instructors in CSE-exposed centres. Sample size of 450 per group was calculated. A structured interviewer-administered questionnaire; focus group discussions and key informant interviews were used for data collection. Quantitative data was analyzed at p<0.05; coding and thematic analysis of qualitative data was followed by integration and of the findings; and interpreted using the information-motivation-behavioural skills model. Students in CSE-exposed schools had statistically significantly higher SRHR-related knowledge scores and positive attitudes towards inclusive gender norms. The CSE training improved the educational knowledge and teaching modalities of the instructors. The adolescents had increased knowledge, acquired behavioural skills, and changed certain behaviors. The SRHR-related benefits extended to their siblings and friends. This study contributes to the knowledge available on the benefits of CSE for young people in non-formal settings in Nigeria. It revealed that CSE training benefits not only the students but also their instructors, who reported acquisition of new teaching skills and improved parenting skills. The study findings suggest that the integration of CSE into non-formal educational settings could improve the SRHR- related knowledge among young people, promote inclusive gender norms and potentially contribute to improved SRH outcomes and related SDGs in Nigeria.

**Data Availability Statement:** Yes, the data excel spreadsheet has been uploaded as supporting document.

**Funding:** The authors received no specific funding for this work.

**Competing interests:** The authors have declared that no competing interests exist.

## Introduction

Adolescents and young people are vulnerable to violence, abuse, exploitation and the harmful health consequences of poor sexual and reproductive health decisions and choices [1–4]. Conservative social norms contribute to a lack of knowledge and awareness about puberty, sexuality, and basic human rights, which have been shown to endanger the health and welfare of young people as well as that of their off springs [5]. Conservative gender norms were identified as drivers of interrelated SRH behaviours including gender-roles and expectations in the following circumstances: contraceptive decision making in Northern Nigeria, sexual permissiveness in eastern Nigeria, female genital mutilation and gender-based violence [6–8]. Comprehensive Sexuality Education (CSE) is a major component of the essential life-saving sexual reproductive health & right program (SRHR) for young people and considered effective in changing gender norms required for improving SRHR-related outcomes [4, 9].

An estimated 22% of Nigeria's 220million population is categorized as adolescents, with the absolute number of Nigerian youths projected to exceed fifty-seven million by 2025 [10]. The gender distribution is estimated at 56% males and 49% females. A large proportion (10.5%) of adolescents in Nigeria participate in some form of informal education described as education that does not take place within the frame of an official curriculum. With regards to the non-formal settings (vocational training, informal religious schools, literacy initiatives), over 70% of the participants are female due to difficulty accessing formal education [11–13].

The factors responsible for the poor health status of adolescents and young people in Nigeria include inadequate access to health information and services, poverty, as well as inequitable gender norms [14]. This is reflected in the data obtained. According to UNICEF, Nigeria has the highest number of children and adolescents aged 0–19 years living with HIV in West and Central Africa, with an estimate of 190,000. In 2020, 4,904 adolescent girls were newly infected with HIV, compared with 1,575 adolescent boys in the country [15]. Furthermore, a scoping review reported that the range of incidence for having sex among unmarried young people in the country varied from 57.2% to 82.7%, and the prevalence of unintended pregnancies from 23.4% to 92.7% [16].

The Family Life and HIV Education (FLHE) curriculum is the school-based intervention for disseminating information about HIV/AIDS, sexually transmitted infections, unintended pregnancies, and inclusive gender norms in Nigeria [17, 18]. However, since the deployment of the curriculum in 2003 for in-school adolescents, their peers in non-formal training settings were left uncatered for.

A systematic review of the effectiveness of HIV/AIDS school-based sexual health education programmes in Nigeria showed that the adolescents had increased knowledge, healthier attitudes and practiced safer sexual health behaviour [19]. Evidence exists that change in behaviour can be attributed to possession of certain modifiable risk factors namely: information, motivation and behavioral skills. Sexual and reproductive health behaviors are acts influenced by these factors [20, 21]. The information-motivation-behavioural risks (IMB) model has been successfully deployed in efforts to understand as well as promote sexual and reproductive health. It specifies the constructs and helps to identify the relationships assumed to exist among the constructs featuring as influencers of sexual and reproductive health practices. The model has been used extensively in promotion of contraceptive use and HIV prevention intervention among adolescents to address information gaps, motivation obstacles and behavioural skills deficits associated with HIV risk exposure [20, 21].

The Lagos State Government with funding from UNFPA-UBRAF (2018 to 2020) developed, launched and integrated the CSE program into non-formal educational settings. The

CSE curriculum and manual has six modules comprising SRHR information and life skills. This study assessed and compared students in CSE exposed and CSE naïve technical and vocational centres in Lagos state to generate a hypothesis regarding its benefits and influence in changing knowledge and attitude. Specifically, to determine and compare comprehensive sexuality education-related knowledge; determine and compare attitudes towards gender norms; and use the IMB model to explore the benefits of teaching and learning about CSE from exposed students and their instructors. The findings will contribute to the body of knowledge regarding this topic among students in non-formal settings and can also be used as an advocacy tool with policy makers.

## Materials and methods

Lagos State is the most densely populated state in Nigeria, with an estimated population of 20,5 million people [22, 23]. The state has a high literacy rate of 92% compared to the national average of 56% [22, 23]. Lagos state has 22 non-formal training institutions (5 technical schools and 17 vocational schools). The courses/trades offered at the vocational centers include Hairdressing & cosmetology; Fashion designing; Textile, Arts & Bead making; Computer studies; Catering; Barbing; Photography. The technical schools offer a wider range of trades such as Bricklaying; Business studies; Furniture making, Plumbing; Welding; Electrical installation; Automobile technology, and Graphic arts.

A mixed-method study design was used made up of a cross-sectional comparative study and a qualitative case study of CSE-exposed technical and vocational centers from 23rd June to 7th July 2022. The study population comprised of male and female students attending technical and vocational centers in Lagos State. To be eligible, the student must be between 15–24 years of age. Sample size was calculated using the Cochran formula $n = z^2pq/d^2$ (n is the minimum sample size, z the standard deviation at 95% confidence interval, p of 0.5 and d of 0.5 was used). Sample size of 384 was obtained which was increased to 500 per group.

Multistage sampling technique was used. In the first stage of sampling, the centers were grouped into CSE naïve and CSE exposed based on whether the students had been taught the CSE curriculum or not. Therefore, we had technical (exposed and naïve); vocational (exposed and naïve). In all there were 13 CSE-exposed centers (technical & vocational) and 9 CSE-naïve centers (technical & vocational). Subsequently, four centers were randomly selected among the 13 CSE-exposed centers and three among the 9 CSE-naïve centers. The sample size for each institution was based on the proportional contribution of that center to the sampling frame population Ten courses were randomly selected from the twelve available in technical centers and five from the seven available in the vocational centers. Students attending those courses were eligible for recruitment into the study. Selection of respondents was done using the school's attendance register.

For the quantitative data collection, trained experienced research assistants (young adults) were used. The questionnaire was adapted from the Lagos State sexuality education questionnaire for young persons and administered via Kobotool software on a programmed mobile device through the Open Data Kit (ODK). As much as practicable, gender matching of respondent to data collector was ensured. Information collected was analyzed using IBM SPSS Statistics version 20 software. Frequency and percentage as well as summary statistics (mean and standard deviation) were generated. The mean was used for the comparison of the two groups at P<0.05. 5 statements were used to assess perception of gender norms using a 3 point Likert scale. The maximum score based on the most appropriate option is 2marks. Maximum score of 10marks. The higher the score, the more appropriate and relevant the attitude. Overall, individuals with scores ≥ 6 are considered to have good attitude.

The Qualitative aspect of the study explored the benefits of teaching and learning CSE among instructors and students in CSE exposed schools, data was collected using three focus group discussions (FGD) with students from 3 schools (10 each) and five key informant interviews with the CSE instructors using FGD/KII guide. The interviews were transcribed verbatim, coded, thematically analyzed and interpreted using the IMB model. 20,21 Qualitative data management tool NVivo was used for data management. The quantitative and qualitative results were integrated for the interpretation of the findings.

Ethical approval for this research was obtained from the Lagos University Teaching Hospital Health Research and Ethics Committee (HREC). Approval was also obtained from the relevant ministries, departments and agencies. Informed consent was obtained from the principals of each of the schools in their capacity as the legal guardian of the students. Participation was voluntary and written informed consent was obtained from all participating students above 18years of age and assent from those below 18years. The questionnaires were anonymous, and respondents were assured of the highest level of confidentiality on information given through appropriate data storage and protection. The instructors and staff of the technical colleges and vocational centers were not directly involved in any aspect of data collection and analysis. Students who decided not to participate were not reported to the instructor/punished, and those who decided to discontinue with the interview at any point during the process were not penalized in any way. Participants' conduct or answers were not discussed with the instructors or staff of the institutions. Data collection took place in the school halls with adequate space to ensure privacy for the participants.

## Results

One thousand students participated in the study however 939 entries (456 from CSE exposed respondents and 483 from CSE naïve respondents) could be analyzed.

### Quantitative analysis

Among the 939 respondents, 547 were females and 392 males. The mean age of the adolescents was 18.14±2.23 for CSE-exposed and 19.24±2.66 for CSE- naïve which was not statistically significant. However, a higher proportion of the exposed were younger than 20 years of age (76.1% vs 61.5%). A statistically significant difference was observed between the CSE-exposed and CSE-naïve students with respect to biological sex (p<0.001); marital status (p = 0.002); level of education (p<0.001); ethnicity (p<0.001) and internet use (p<0.001). A higher proportion of the exposed were male (50.7% vs 33.3%); single (98.2% vs 94.4%); were less educated (59.6% vs 12.2%); and of Igbo ethnic group (28.7% vs 18.6%). However, a higher proportion of naïve students engage with the internet on a daily basis (81.4% vs 62.3%).

**Knowledge of reproductive organs.** With regards to CSE-related knowledge, less than a third of all the students (30.9%) knew the seven reproductive organs asked. A higher proportion of the knowledgeable students were in the CSE-exposed schools (35.7% vs 26.3%; p<0.001). More than half had above average knowledge of the reproductive organs (57.2%) with a higher proportion among the exposed (61.4% vs 53.2%; p = 0.038). The mean knowledge score was statistically significantly higher in the CSE-exposed group (5.16±1.776 vs 4.66 ±1.943; p<0.001). Students were highly knowledgeable of the vagina and penis and less knowledgeable of the clitoris and scrotum. A statistically significant difference was revealed between the exposed and naïve students on the knowledge of five organs namely: Testes; Penis: Scrotum: Uterus and Ovaries while the knowledge of the Vagina and Clitoris was similar.

**Knowledge of condom and contraceptives.** Similar proportion of exposed and naïve students had seen male condoms (76.3% vs 73.9%) and female condoms (24.0% vs 20.2%).

Students in CSE-exposed schools were more knowledgeable about the role of condoms in preventing pregnancy (86.8% vs77.2%; p<0.001); preventing HIV transmission (81.4% vs 73.1%; p = 0.002); preventing other STIs (83.1% vs 71.8% p<0.001). Overall, two-thirds (68.1%) of the students had good knowledge, the majority of whom were in the CSE-exposed schools (74.6% vs 62.1%). The mean knowledge score was statistically significantly higher in CSE-exposed students (2.51±0.965 vs 2.25±1.123; p<0.001). Knowledge of contraceptives was very poor, 5.0% and 1.0% of the exposed and naïve students knew all the following modern contraceptives: Pills, Emergency contraceptives, Injectable contraceptives, Implants, and IUCD. The least popular for the two groups was IUCD (13.4% vs 10.8%). There was a statistically significant difference in knowledge of pills (68.9% vs 61.0% p 0.007); emergency contraceptives (29.3% vs 11.4%; p<0.001); injectable (43.8% vs 21.4% p 0.000) and implants (25.3% vs 16.4% p 0.001). The mean knowledge score was 1.77±1.472 vs 1.20±1.252; p<0.001 Table 1.

**Knowledge of HIV and PMTCT.** Overall, about 1% of the students had correct knowledge of all the ten items used to assess HIV transmission and prevention, with a higher proportion in CSE-exposed schools (1.1% vs 0.6% p<0.001). The mean HIV knowledge score was 6.70±1.474 vs 6.34±1.607, p<0.001. Knowledge of HIV transmission via unprotected sex was similar for the two groups and was the highest among the transmission-related questions. Myths and misconceptions regarding HIV transmission were high in both groups with 91.4% of CSE-exposed students and 96.8% of CSE-naïve students saying that HIV can be transmitted via hugging; a further 37.6% and 29.6% sharing of toilets while 78.1% and 83.0% knew that antibiotics can prevent HIV transmission. A statistically significant difference was observed with regards to PMTCT-related mean knowledge score (2.46±1.044 vs 1.92±1.005; p<0.001) and overall good knowledge (43.6% vs 28.2% p<0.001). PMTCT-related knowledge was lowest

**Table 1. Knowledge of condom and contraceptives.**

| Variable | CSE Exposed | CSE Naive | All | |
|---|---|---|---|---|
| | Freq (%) | Freq (%) | Freq (%) | $X^2$; p value |
| **Condom** | | | | |
| Have seen male condoms | 348 (76.3) | 357 (73.9) | 705 (75.1) | 0.72; 0.219 |
| Have seen female condoms | 16 (24.0) | 92 (20.2) | 208 (22.2) | 2.01; 0.090 |
| Condom prevents unplanned pregnancy | 396 (86.8) | 373 (77.2) | 769 (81.9) | 14.63; 0.000 |
| Condom prevents HIV transmission | 371 (81.4) | 353 (73.1) | 724 (77.1) | 9.10; 0.002 |
| Condom prevents other STI | 379 (83.1) | 347 (71.8) | 726 (77.3) | 16.99; 0.000 |
| Overall (knew all the 3 answers) | 340 (74.6) | 300 (62.1) | 640 (68.1) | 74.85; 0.000 |
| Mean knowledge score | 2.51±0.965 | 2.25±1.123 | | t = 3.729 p = 0.000 CI 0.123–0.398 |
| **Modern contraceptive methods** | | | | |
| Pills | 308 (68.9) | 294 (61.0) | 602 (64.8) | 6.36; 0.007 |
| Emergency contraceptives | 131 (29.3) | 55 (11.4) | 186 (20.0) | 46.38; 0.000 |
| Injectable | 196 (43.8) | 103 (21.4) | 299 (32.2) | 53.69; 0.000 |
| Implants | 113 (25.3) | 79 (16.4) | 192 (20.7) | 11.18; 0.001 |
| IUCD | 60 (13.4) | 52 (10.8) | 112 (12.1) | 1.52; 0.129 |
| Overall (knew all the 5 methods) | 23 (5.0) | 5 (1.0) | 28 (3.0) | 43.33; df 4; p 0.000 |
| Good knowledge (knew at least 3 or more methods) | 137 (30.0) | 81 (16.8) | 218 (23.2) | 23.18; 0.000 |
| Mean knowledge score | 1.77±1.472 | 1.20±1.252 | | t = 6.221 p = 0.000 CI 0.394–0.757 |

**Table 2. Knowledge of HIV and PMTCT.**

| Variable | CSE Exposed | CSE Naive | All | |
|---|---|---|---|---|
| | Freq (%) | Freq (%) | Freq (%) | $X^2$; p value |
| **HIV transmission** | | | | |
| Unprotected sex | 435 (98.0) | 463 (97.3) | 898 (97.6) | 0.49; 0.316 |
| Blood transfusion | 416 (93.7) | 412 (86.6) | 828 (90.0) | 13.01; 0.000 |
| Sharing toilets | 167 (37.6) | 141 (29.6) | 308 (33.5) | 6.59; 0.006 |
| Sharing sharp objects | 427 (96.2) | 426 (89.5) | 853 (92.7) | 15.16; 0.000 |
| Hugging | 406 (91.4) | 461 (96.8) | 867 (94.2) | 12.37; 0.000 |
| **HIV prevention** | | | | |
| Having one uninfected faithful partner | 340 (74.6) | 358 (74.3) | 698 (74.4) | 0.10; 0.490 |
| Consistent condom use | 373 (81.8) | 383 (79.5) | 756 (80.6) | 0.82; 0.206 |
| Abstinence | 371 (81.4) | 396 (82.2) | 767 (81.8) | 0.10; 0.408 |
| Using antibiotics | 356 (78.1) | 400 (83.0) | 756 (80.6) | 3.62; 0.034 |
| Avoid sharing sharps | 391 (85.7) | 384 (79.7) | 775 (82.6) | 6.03; 0.009 |
| Overall (knew all the 10 answers) | 5 (1.1) | 3 (0.6) | 8 (0.9) | 30.29; df 9; 0.000 |
| Overall (knew at least 6 of the 10 answers) | 388 (85.1) | 353 (73.1) | 741 (78.9) | 20.31; 0.000 |
| Mean knowledge score | 6.70±1.474 | 6.34±1.607 | | t = 3.705 p = 0.000 CI 0.169–0.550 |
| **PMTCT** **HIV can be transmitted from mother to child** | | | | |
| During pregnancy | 264 (59.1) | 179 (37.6) | 443 (48.0) | 42.52; 0.000 |
| During childbirth | 204 (45.6) | 163 (34.2) | 367 (39.8) | 12.49 0.000 |
| Through breastfeeding | 324 (72.5) | 285 (59.9) | 609 (66.0) | 16.33; 0.000 |
| ARVs prevent MTCT of HIV | 308 (69.1) | 298 (62.6) | 606 (65.7) | 4.26; 0.023 |
| Overall (knew all the 4 answers) | 90 (19.7) | 32 (6.6) | 122 (13.0) | 66.79; 0.000 |
| Overall (knew at least 3 of the 4 answers = good knowledge) | 199 (43.6) | 136 (28.2) | 335 (35.7) | 24.50; 0.000 |
| Mean knowledge score | 2.46±1.044 | 1.92±1.005 | | t = 7.666 p = 0.000 CI 0.400–0.067 |

on childbirth-related HIV transmission (45.6% vs 34.2%; p<0.001) but highest on the use of ARVs (69.1% vs 62.6%; p 0.023) Table 2.

**Attitudes towards gender norms.** This was generally negative, overall, 13.1% had positive attitudes towards inclusive gender norms. In the CSE naïve group, 53 (10.9%) of the students had good attitudes, however, the gender-based difference observed was not statistically significant. Also, in the CSE exposed group, 70(15.4%) of the students had good attitudes towards gender norms. The gender-based difference observed within this group was not statistically significant as well. Intragroup comparison of mean attitude score shows higher scores among female students compared to their male counterparts in the naïve and exposed group which was statistically significant for each group. The gender-based difference observed in the naïve group with regards to two statements namely: Some house chores are specifically for males and some for females (p = 0.009 vs 0.225) and Husband & wife should contribute to family income (p = 0.002 vs 0.524) was erased in the exposed group. Both groups did not exhibit gender-based difference with regards to whether men and women should be given the same opportunities. Across board, negative attitude was exhibited towards this statement. The following statements: Men are usually better than women in school and the workplace & Families should spend equally on the education of daughters

**Table 3. Attitude towards gender norms among CSE naïve students.**

| Variable | Female | Male | Female | Male | Female | Male | |
|---|---|---|---|---|---|---|---|
| Gender norms | Agree n (%) | | Indifferent n % | | Disagree n (%) | | X²; p |
| Some house chores are specifically for males and some for females | 167 (52.2) | 107 (65.6) | 13 (4.1) | 08 (4.9) | 140 (43.8) | 48 (29.5) | 9.29; 0.009 |
| Men are usually better than women in school and the workplace | 39 (12.2) | 75 (46.0) | 31 (9.7) | 22 (13.5) | 250 (78.1) | 66 (40.5) | 77.15; 0.000 |
| Men and women should be given the same opportunities | 17 (5.3) | 12 (7.36) | 7 (2.2) | 03 (1.8) | 296 (92.5) | 148 (90.8) | 0.85; 0.653 |
| Husband & wife should contribute to family income | 59 (18.4) | 21 (12.9) | 43 (13.4) | 08 (4.9) | 218 (68.1) | 134 (82.2) | 12.39; 0.002 |
| Families should spend equally on the education of daughters and sons | 14 (4.4) | 16 (9.8) | 10 (3.1) | 9 (5.5) | 296 (92.5) | 138 (84.7) | 7.46; 0.024 |

Grading

The desirable response is 2marks, Indifferent is 1mark, undesirable response 0mark

Maximum score of 10 (scores $\geq$ 6 = positive attitude)

Female 39 (12.2) Male 14 (8.6) Total 53 (10.9) $X^2$ 1.43; p 0.231

Mean score Female 3.33±1.71

Mean score Male 2.31±1.93

t = -5.92; p 0.0000; CI -1.357 to—0.680

**Table 4. Attitude towards gender norms among CSE exposed students.**

| Variable | Female | Male | Female | Male | Female | Male | |
|---|---|---|---|---|---|---|---|
| Gender norms | Agree n (%) | | Indifferent n % | | Disagree n (%) | | X²; p |
| Some house chores are specifically for males and some for females | 133 (58.6) | 152 (66.4) | 09 (3.9) | 08 (3.5) | 85 (37.4) | 89 (30.1) | 2.98; 0.225 |
| Men are usually better than women in school and the workplace | 29 (12.8) | 90 (39.3) | 05 (2.2) | 22 (9.6) | 193 (85.0) | 117 (51.1) | 60.59; 0.000 |
| Men and women should be given the same opportunities | 36 (15.9) | 32 (13.9) | 4 (1.8) | 03 (1.3) | 178 (82.4) | 194 (84.7) | 0.49; 0.779 |
| Husband & wife should contribute to family income | 55 (24.2) | 54 (23.6) | 09 (3.9) | 05 (2.2) | 163 (71.8) | 170 (74.2) | 1.29; 0.524 |
| Families should spend equally on the education of daughters and sons | 16 (7.05) | 31 (13.5) | 01 (0.44) | 09 (3.9) | 210 (95.5) | 189 (82.5) | 12.28;0.002 |

Grading

The desirable response is 2marks, Indifferent is 1mark, undesirable response 0mark

Maximum score of 10 (scores $\geq$ 6 = positive attitude)

Female 36 (15.9) Male 34 (14.9) Total 70 (15.4) $X^2$ 0.089; p 0.764

Mean score Female 3.52±1.70

Mean score Male 2.85±2.07

t = -3.73; p 0.0002; CI -1.013 to -0.315

and sons elicited statistically significant gender-based difference in attitude from the CSE exposed and naïve students Tables 3 & 4.

## Qualitative analysis

**Theme: Adolescents and instructors' perception of training-related outcomes achieved.** At the individual level, most of the adolescents and their instructors mentioned a range of benefits they obtained from attending the CSE training, this cuts across 3 domains namely knowledge learned/gained; behavioural skills acquired and behaviour change. Learning correct information increases the knowledge of students thereby correcting misinformation (learn unlearn & relearn). Young people were amazed to learn that they have rights. Learners were also of the opinion that some of the content was not entirely new because they had been exposed to some of the topics during earlier school attendance; however, the method of teaching the CSE curriculum improved their understanding of a broad range of topics. The

adolescents were of the opinion that they need the information, and the life skills taught to navigate successfully into adulthood Fig 1.

*I learnt that using abusive words to your fellow teammates or friend to one another is not good and it can cause violence and since I have been to the lecture on gender-based violence, and I practiced it, I can see that it improved my life.*

**(16 years, Male, SSS, Computer**)

**Knowledge Gained**
- **Student**
- learnt how to relate with people
- learnt how to communicate effectively
- knows contraceptive methods
- knows about puberty & menstral hygiene
- knows rights
- correction of misconceptions
- knows about adolescent body (male & female)
- **Instructor**
- improved educational knowledge

**Skills Acquired**
- **Student**
- develop ideas to prevent sexual harrassment
- defend self in the event of sexual abuse
- plan & execute plans for the future
- self control when on a date
- self-confident & bold & outspoken
- set goals
- self control
- be creative
- ability to make & execute decisions
- communicate effectively
- **Instructor**
- how to relate, interact with and teach the students the way the curriculum was designed to be taught

**Behaviour Change**
- **Student**
- does not engage in verbal abuse anymore
- proper conduct with the opposite sex
- stopped sharing shrap objects with friends
- improvement in lifestyle
- **Instructor**
- adoption of the CSE teaching method for core courses
- improved interaction with their children
- helping colleagues with CSE related issues

**Fig 1. Self-reported knowledge, skills and behaviours among instructors and CSE-exposed students.**

*It was eye opening because most things we have the knowledge, but few were lies or wrong information's. So, getting to go through the process make us understand more about the topic and subject we were taught.*

**(21years, Female, SSS, Computer)**

*What has really changed in my life is that I learnt more about rights. Like now maybe some-one brutalize you and you know what to do. Some people might be scared because he is older than me, I don't have to tell anybody what he did to me but since I have been to the lecture I realized that there is nothing anybody can do to me in this world. Since you have a right, you have the rights to do anything, maybe report to the police or human rights. So, what has changed in me is that I have known my rights.*

**(21years, male, SSS, Computer)**

*Based on this subject well, I had, I now know how to defend myself when I want to be sexually abused, and I also know how to set my long- and short-term goals that I aim to achieve.*

**(16 years, Female, JSS, Computer)**

*There are many methods of teaching one can just come to the class and display a playlet or one can come up with some materials just display the materials and decide to talk with those materials from those materials your topic will come out even you may not write the topic at the initial stage but when you are commending or saying something on the materials you have pasted from there the audience will deduce your topic.*

**(34years, female instructor)**

The instructors reported that the CSE curriculum and its training of the trainer's work-shop built the capacity of the instructors through improved educational knowledge of the topics & the acquisition of new skills in teaching modalities. Other benefits mentioned include ability to relate, interact with and teach the students the way the curriculum was designed to be taught. Seeing how successful the course was in bringing out the best in students while learning, some of the instructors deployed the method in teaching the core courses and in their interactions with students generally and their children at home. The instructors believed that being a CSE instructor impacted positively on their teaching and parenting skills Fig 1.

*. . . . . . .. So it has been helping me in classes in even teaching my own subject trying to be lenient in class not going with that straight face that I am here to teach and you must listen but the way we do it there that you laugh, play to make the class interesting yes I have learnt that and am also using it to teach my normal Biology apart from the Sexuality Education.*

**(46years, female instructor)**

*. . . . . . .. it increased my knowledge in the area of Sexuality Education then it gives me this opportunity to be able to relate with children . . . . . . I tend to have more of them coming closer because they too find out that this man has a listening ear so we can move in fact things that they cannot even tell their teachers they do come around and say I want to see you oooh. I just need to ask questions and you know just like that so I tend to be closer to them than before.*

**(46years, male instructor)**

According to the adolescents, the benefits from the training extended to their siblings and friends. Improved communication skills obtained from the training empowered them to talk and relate easily with people in their immediate environment on CSE-related topics. Having learned the correct information, the adolescents were motivated to give useful and beneficial advice to friends and siblings in areas such as: sexual relationships (how to protect themselves from STI during sex, refusing unwanted sexual intimacy, and sexual harassment). They were also able to provide information on self-esteem and values (to help them know their worth in everything they do); how to be bold; how to start their own business to their peers Fig 1.

*Well it has helped me to talk to my friends better like before I was not really assertive with my words like I could not really tell people what is in my mind and all those things but now I talk to people and tell them that okay this is what you should do this is what seems right without manipulating them like I've helped them to think properly.*

*(18 years, Female, SSS, Fashion)*

*The reason why I like the topic on dating is that it helps you to come up with the agreement between you and your friend that want to go for dating like this is what we are going to do, this is what we are not going to do like this is where you are going to touch this is where you are not going touch in my parts you understand.*

*(22years, Male, JSS, Block laying)*

*It helps me to be more bold and more eloquent in my speech, in my behaviours and I know my worth now. I know what I am supposed to do and what am not supposed to do to prevent some certain harassment and abuse.*

*(17years, Female, Post SSS, Fashion)*

## Discussion

This cross-sectional study carried out among adolescents and young people in non-formal educational settings in Lagos, Nigeria demonstrated significant differences in knowledge of SRHR issues and gender-based differences in gender norms based on exposure to the CSE curriculum. Furthermore, this was supported by self-reported changes in knowledge; behaviour skills, adoption of behaviours/lifestyles promoting SRHR and inclusive gender norms by students and their instructors in the CSE exposed centers. We applied the IMB model to improve our understanding of the relationships existing between the CSE curriculum as a tool for teaching and learning by instructors; vulnerable adolescents and young people in non-formal educational settings. We were able to highlight potential linkages between the SRHR information in the modules, the SRH knowledge acquired and the SRH motivation developed by the adolescents; the connection between these constructs and the life skills component of the curriculum; as well as the potential pathway for the SRH positive behaviours adopted by the adolescents. The topics taught in the modules provided information on several SRHR issues leading to knowledge acquisition while at the same time building their SRH related behavioural skills capacity using life skills. The students were sufficiently motivated to adopt safer SRH related behaviours such as HIV/AIDS, STI and pregnancy prevention voluntarily (Fig 2). This process was encouraged by the andragogy form of delivering the modules which is facilitation based compared to the pedagogy enshrined in the routine teaching of most courses in

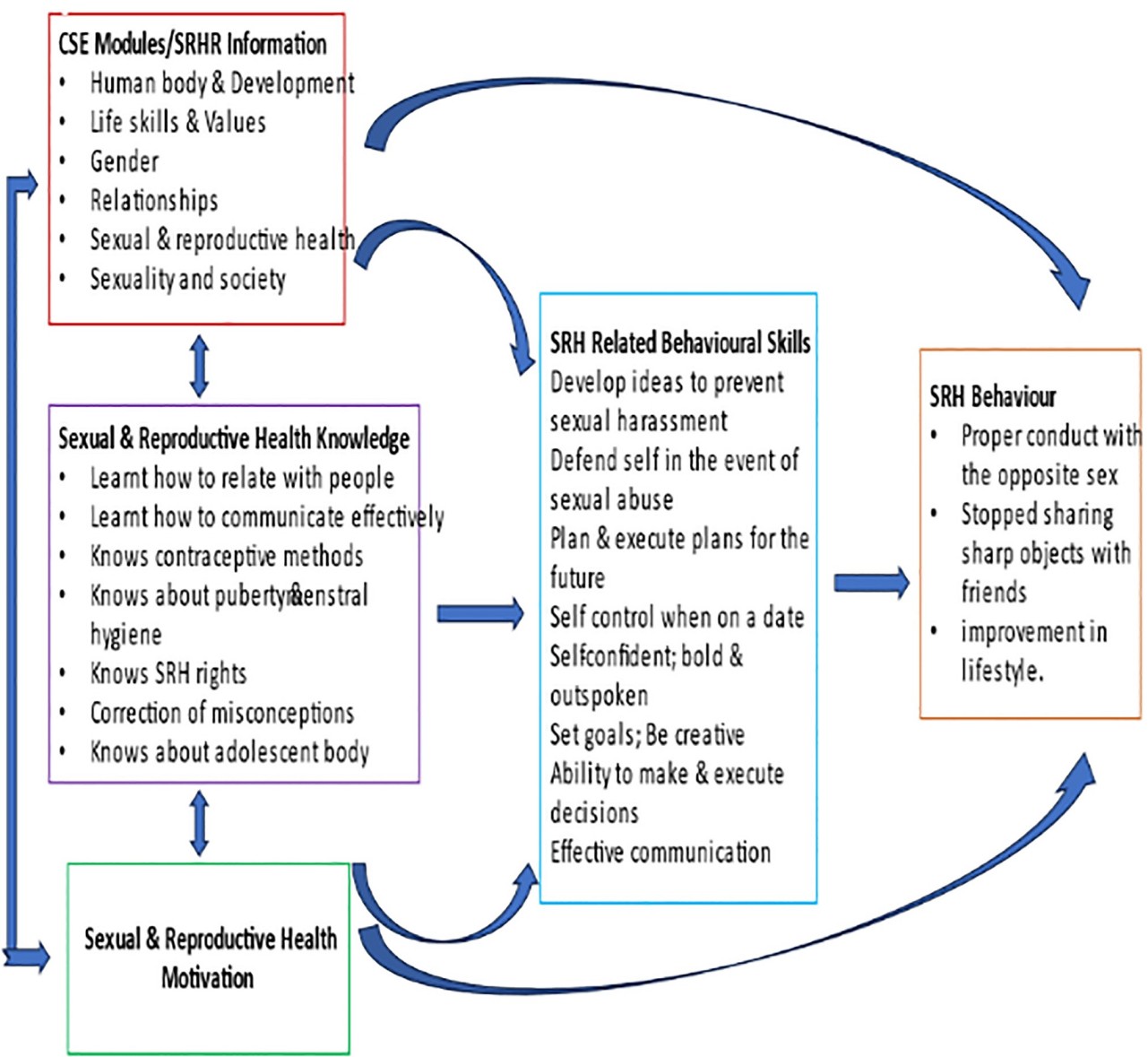

**Fig 2. Adapted information-motivation-behavioural skills model for CSE sexual reproductive health & rights behaviour.**

the non-formal settings. Based on the experiences of the instructors in this study with this method of adult education knowledge delivery and its application to other courses outside of the CSE curriculum andragogy method appears suitably matched to meet the needs of the instructors and the learners in the non-formal setting studied.

With an estimated population of 10 million out-of-school adolescents, [24] research and publications focusing on CSE for this population in Nigeria is limited. Only one third of all the adolescents knew of the following reproductive organs; clitoris; vagina; penis; testes; uterus; ovaries and scrotum. This might be due to the cultural environment (conversations around sexuality related topics is considered a taboo), notwithstanding, the CSE-exposed adolescents had higher scores. This is in line with existing evidence on the ability of CSE delivered in out-

of-school settings to provide the necessary information required by young people to understand their bodies and issues around body autonomy [25, 26].

Comprehensive knowledge of HIV prevention was very poor among our study population. This is in tandem with the findings of a previous study from a nationally representative sample of young persons in the country [27] and suggest that young people have been left behind in HIV programming. Although a higher proportion of the CSE-exposed population had good knowledge (1.1% vs 0.6%) and were more knowledgeable, this falls short of the 11.6% of women and 7.0% of men (15–24 years) with comprehensive knowledge recorded in Lagos by the NDHS 2018 [28]. Poor HIV-related knowledge has been linked to poor practice and risky sexual behaviour which increases susceptibility to infection with STI including HIV [29–31]. The postCOVID19 pandemic observation of an estimated one third of new infections occurring among young people therefore requires CSE-based interventions. PMTCT related knowledge mirrored HIV-related knowledge, this is expected because evidence exist that knowledge of both items is usually unidirectional. Population groups with high comprehensive knowledge of HIV will also possess high PMTCT-related knowledge, [32, 33] which is considered vital in preventing MTCT of HIV [34, 35].

In terms of knowledge of condoms and contraceptives, the two groups were similar, overall, 7 in 10 adolescents had seen a male condom compared to 2 in 10 for female condoms. However, students in CSE-exposed schools compared to CSE-native schools had greater knowledge about the role of condoms in preventing pregnancy, HIV transmission and other STIs. Although knowledge of modern contraceptives was generally poor (30% vs 16.8%), the proportion of students in CSE-exposed schools who had good knowledge of modern contraceptives was significantly higher. This is lower than the 72% recorded in a baseline survey of street-involved young people (SIYP) aged 10–24 years in Lagos and Osun state, Nigeria [35]. Furthermore, studies carried out among adolescents in Nigeria has consistently reported poor knowledge about reproductive health and practices, misconceptions about contraceptives and poor sexuality education [36, 37].

This study, which is one of the few studies in Nigeria to assess the effect of CSE on gender norms, found that only 13.1% of all participants (statistically higher in the CSE-exposed group) exhibited positive attitudes towards inclusive gender norms. Support for gender equality in educational opportunities was offered by a tenth of the CSE-exposed group. This contrasts with findings from a multi-country study in East and Southern Africa, where CSE interventions led to more equitable gender attitudes related to education [38]. However, the majority were supportive of non-gendered household chores which mirrors global sentiments as reported by a systematic review which found that school-based interventions, including CSE, had a moderate impact on improving gender-equitable attitudes, including those related to domestic roles [39]. At the same time, it resonates with a global review by UNESCO, which highlighted the positive impact of quality CSE on gender attitudes and roles and underscore the potential of CSE in promoting gender equity, even in varied cultural contexts [40].

## Conclusion

There are a limited number of studies assessing benefits of CSE-training for students in non-formal settings. The findings of this study therefore contribute to the body of knowledge on the benefits of CSE for out-of-school young persons and their instructors in SSA. However, in the absence of baseline measurement, it is impossible to categorically state that the CSE curriculum was solely responsible for the difference observed between the two groups of adolescents in our study. Regardless, by demonstrating consistently improved SRHR outcomes, the CSE curriculum has the potential to advance the well-being of adolescents and young adults in

non-formal settings in other states within the country as well as countries with similar backgrounds.

This research provided a window into how educational interventions can shape perspectives among out-of-school adolescents, young individuals, and instructors in skills-based non-formal settings. When contextualized with broader research, the findings offer both hope and direction for future initiatives aimed at achieving Sustainable Development Goals 3,5 and 10. However, given that there were clear areas of weaknesses in SRHR knowledge and gender-based differences were observed among the CSE-exposed students in this study, a review of the curriculum and teaching methodology is required to identify areas for improvement to increase its effectiveness.

This study generated a hypothesis regarding the ability of the CSE curriculum to make a difference in knowledge, attitude towards inclusive social norms, acquisition of behaviour skills and adoption of positive SRHR behaviours. However, the limitation of this study design is that it does not allow controlled assessment of expected study outcomes both at the baseline and endline (attributable difference) and the influence of compounding factors. The authors recommend randomized control trials (RCT) to evaluate the effectiveness of the CSE curriculum and cost benefit analysis to compare CSE with other HIV related programming activities targeted at adolescents and young people in non-formal settings.

In conclusion, our study offers a snapshot of the transformative potential of CSE in non-formal settings. When viewed alongside broader research from the African continent and globally CSE remains a powerful tool in promoting the wellbeing of adolescents and young persons.

## Supporting information

**S1 Data.**
(XLSX)

**S1 Checklist. STROBE checklist.**
(DOCX)

## Author Contributions

**Conceptualization:** Babatunde ADELEKAN, Esther SOMEFUN, Olushola KAREEM, Oladimeji IBRAHIM, Hamira WELYE, Rabiatu SAGEER, Erika GOLDSON, Rashidat UMAR, Adekemi Oluwayemisi SEKONI.

**Data curation:** Babatunde ADELEKAN, Esther SOMEFUN, Olushola KAREEM, Oladimeji IBRAHIM, Hamira WELYE, Adekemi Oluwayemisi SEKONI.

**Formal analysis:** Babatunde ADELEKAN, Esther SOMEFUN, Olushola KAREEM, Hamira WELYE, Adekemi Oluwayemisi SEKONI.

**Funding acquisition:** Babatunde ADELEKAN, Esther SOMEFUN, Hamira WELYE, Erika GOLDSON.

**Investigation:** Babatunde ADELEKAN, Esther SOMEFUN, Olushola KAREEM, Oladimeji IBRAHIM, Hamira WELYE, Rashidat UMAR, Adekemi Oluwayemisi SEKONI.

**Methodology:** Babatunde ADELEKAN, Esther SOMEFUN, Olushola KAREEM, Oladimeji IBRAHIM, Hamira WELYE, Rabiatu SAGEER, Rashidat UMAR, Adekemi Oluwayemisi SEKONI.

**Project administration:** Babatunde ADELEKAN, Esther SOMEFUN, Olushola KAREEM, Oladimeji IBRAHIM, Hamira WELYE, Rabiatu SAGEER, Erika GOLDSON, Rashidat UMAR, Adekemi Oluwayemisi SEKONI.

**Resources:** Babatunde ADELEKAN, Esther SOMEFUN, Olushola KAREEM, Oladimeji IBRAHIM, Hamira WELYE, Rabiatu SAGEER, Erika GOLDSON, Adekemi Oluwayemisi SEKONI.

**Software:** Esther SOMEFUN, Olushola KAREEM, Adekemi Oluwayemisi SEKONI.

**Supervision:** Babatunde ADELEKAN, Esther SOMEFUN, Olushola KAREEM, Oladimeji IBRAHIM, Hamira WELYE, Rabiatu SAGEER, Erika GOLDSON, Rashidat UMAR, Adekemi Oluwayemisi SEKONI.

**Validation:** Babatunde ADELEKAN, Esther SOMEFUN, Olushola KAREEM, Oladimeji IBRAHIM, Hamira WELYE, Adekemi Oluwayemisi SEKONI.

**Visualization:** Babatunde ADELEKAN, Esther SOMEFUN, Olushola KAREEM, Adekemi Oluwayemisi SEKONI.

**Writing – original draft:** Esther SOMEFUN, Olushola KAREEM, Adekemi Oluwayemisi SEKONI.

**Writing – review & editing:** Babatunde ADELEKAN, Esther SOMEFUN, Olushola KAREEM, Oladimeji IBRAHIM, Hamira WELYE, Rabiatu SAGEER, Erika GOLDSON, Rashidat UMAR, Adekemi Oluwayemisi SEKONI.

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
