## [Decision Letter · Decision Letter 0]

26 Jun 2024

PGPH-D-24-00960

Comparison of comprehensive sexuality education knowledge and attitudes in CSE-exposed and CSE-naïve non-formal settings in Lagos State, Nigeria.

Dear Dr. sekoni,

Thank you for submitting your manuscript to PLOS Global Public Health. After careful consideration, we feel that it has merit but does not fully meet PLOS Global Public Health’s publication criteria as it currently stands. Therefore, we invite you to submit a revised version of the manuscript that addresses the points raised during the review process.

EDITOR: Please carefully address the reviewer's comments by providing point-to-point response to each comment. Please read the instructions below:  

We look forward to receiving your revised manuscript.

Kind regards,

Tanmay Bagade, Ph.D., MS (O&G), MPH, MHM

Academic Editor

Journal Requirements:

1. Please send a completed 'Competing Interests' statement, including any COIs declared by your co-authors. If you have no competing interests to declare, please state "The authors have declared that no competing interests exist". Otherwise please declare all competing interests beginning with the statement "I have read the journal's policy and the authors of this manuscript have the following competing interests:"

3. Please provide separate figure files in .tif or .eps format only and remove any figures embedded in your manuscript file. Please also ensure all files are under our size limit of 10MB.

4. We have noticed that you have uploaded Supporting Information files, but you have not included a list of legends. Please add a full list of legends for your Supporting Information files after the references list.

Additional Editor Comments (if provided):

Reviewers' comments:

Reviewer's Responses to Questions

**Comments to the Author**

1. Does this manuscript meet PLOS Global Public Health’s publication criteria? Is the manuscript technically sound, and do the data support the conclusions? The manuscript must describe methodologically and ethically rigorous research with conclusions that are appropriately drawn based on the data presented.

Reviewer #1: Partly

Reviewer #2: Partly

2. Has the statistical analysis been performed appropriately and rigorously?

Reviewer #1: Yes

Reviewer #2: Yes

3. Have the authors made all data underlying the findings in their manuscript fully available (please refer to the Data Availability Statement at the start of the manuscript PDF file)?

Reviewer #1: Yes

Reviewer #2: Yes

4. Is the manuscript presented in an intelligible fashion and written in standard English?

Reviewer #1: Yes

Reviewer #2: Yes

5. Review Comments to the Author

Reviewer #1: Review points for PGPH-D-24-00960

Introduction

Need additional context information including;

1. Please provide detailed information about ‘conservative social norms’ particularly ‘gender norms’ in Nigeria, and its effects on sexuality and reproductive health rights.

2. Please provide demographic information about gender distribution of Nigerian adolescent, and by formal and non-formal educational settings.

3. Please provide HIV/AIDs & STDs epidemiological and unintended pregnancies information among adolescent population in Nigeria with arguments related to gender norms and CSE.

Material and Methods

Among 22 non-formal settings including 5 technical (having 12 courses) and 17 vocational centers (having 7 courses),

4. please better clarify the sampling methods to get 10 courses of technical centers and 5 courses of vocational centers;

5. and how to recruit 13 CSE-exposed courses and 9 CSE-naïve courses;

6. and how to recruit CSE-exposed respondents and CSE-naïve respondents, which come out totally 939 respondents for cross-sectional survey.

7. Then state the sub-total numbers of CSE-exposed respondents and CSE-naïve respondents in the starting line of quantitative results.

8. Please also briefly state the scope of issues to be interviewed in FGD in method section.

9. Please recheck the explanation of 3-point Likert scale for assessing perception of gender norms, if it is correct that the maximum score based on the most appropriate option is 2 marks.

Results

The qualitative findings need more supportive evidences (quotations) of benefits of CSE, for the following stated issues;

10. Correcting misconception or misunderstanding

11. The sexuality and reproductive health rights that never known before

12. Teaching modalities that help improving understanding

13. Information/skills that help to navigate successful adulthood

Discussion

14. Overall comment, please start with the similarity and different results between exposed and naïve groups and address particular outcomes in relation to explain gender norm inequality.

15. Please use IMB model to explain the causal relationship towards transforming negative behaviors regarded gender norms to improve sexuality and reproductive health by CSE such as behavior to prevent HIV/AIDs & STD transmission and infection, unintended pregnancies.

16. Please point out some particular finding issues to tackle for further implementation CSE in non-formal educational settings.

17. Please discuss household or domestic gender inequality that influence sexuality and reproductive health.

18. Please discuss the issue of pedagogy and andragogy in term of adult learning for non-formal setting.

Conclusion

19. Please conclude specific findings that support scaling up access to CSE for adolescent in non-formal setting.

Reviewer #2: The main objective of the study was to compare the sexual reproductive health and rights related knowledge and attitudes towards inclusive gender norms among students in comprehensive sexuality education (CSE) exposed and naïve technical and vocational centers in Lagos state, Nigeria. It also explored the benefits of teaching and learning about CSE from CSE-exposed students and their instructors.

The strength of the manuscript is to shed the light to the knowledge and attitudes related to sexual and reproductive health, including gender norms among students in technical and vocational centers in Lagos state, Nigeria. These findings could help to inform improvement of CSE curriculum as well as implementation of other interventions to address the gap in non-formal educational settings in the Lagos state, Nigeria. The manuscript also compares the knowledge and attitudes of CSE in intervention and non-intervention centers at the endline, however, the cross-sectional study design, as authors rightly note, is not an ideal design to demonstrate improvement, attributable to intervention; It does not explore potential compounding factors, including those in which CSE exposed and CSE-Naïve student were statistically different (such as sex, marital status and level of education) and does not allow controlled assessment of expected study outcomes before and after the intervention (attributable difference).

Below, please find key suggestions to improve the manuscript. The lines of the manuscript are not numbered per PLOS LaTeX template. It is therefore difficult to provide detailed review of the manuscript. In the review, I will therefore highlight the feedback per sections without indicating lines

Abstract

1. Based on the focus group interviews and key informant interviews in CSE-exposed centers, the study concludes that “CSE training benefits not only the students but also their instructors, who acquire new teaching skills and improve their parenting skills”. It is important to highlight that these claims are made based on self-reported data.

2. While the conclusion about importance to integrate CSE into non-formal education settings could be made, the claim that “CSE-exposed schools had statistically significantly higher SRHR-related knowledge scores and positive attitudes towards inclusive gender norms” should be revised to clarify that the difference in knowledge and attitudes among CES exposed and naïve centers, although significant, was only measured at the endline and could not be attributed to the intervention without the knowledge of baseline data and students’ previous exposure to CSE curriculum in schools.

3. Similarly, without the clear evidence, the abstract conclusion should be revised as: “The study findings suggest that the integration of CSE into non-formal educational settings could improve SRHR-related knowledge among young people, promote inclusive gender norms and contribute to improve SRH outcomes and related SDGs in Nigeria”.

Introduction

1. The manuscript describes the evidence related to The Family Life and HIV Education (FLHE) curriculum is the school-based settings in Nigeria and related systematic review, that highlights the effectiveness of FLHE program in improving knowledge, attitude and safer sexual health behaviors among youths. For actionability and scale up, it would be helpful to compare the content of the FLSE curriculum to integrated CSE program (intervention) for non-formal education settings. Furthermore, comparative description of these two curricula (key themes) and the six modules CSE program would help to better understand the program results on one hand and identify actionable recommendations to further improve the curriculum in the areas, where CSE-exposed students did not show improved knowledge and/or attitude.

Materials and Methods:

1. Cross-sectional study design, as authors rightly note, is not an ideal design to demonstrate improvement, attributable to intervention, as it does not allow controlled assessment of expected study outcomes both at the baseline and endline (attributable difference) and the influence of compounding factors.

2. Please, provide further information on the composition of focus groups and key informant interviews, particularly their representativeness (number/% of selected CSE study cites) and the method of selection of participants for GFD and KIIs

Results

Quantitative Analysis

1. Given statistically significant difference in sex of study participants between CSE and non-CSE sites, it would be critical to disaggregate study findings by sex and present sex-disaggregated findings for gender norms in two tables (one for boys and one for girls), instead of Table 3 and move Table 3 in the annexes.

Qualitative Analysis

1. The reflection from Female JSS needs to be checked for consistency. It now reads as” based on this subject well, I had, I now know how to defend myself when I want to be sexually abused…”.

2. Please, present the KIIs results on page 12 as self-reported claims by instructors (last paragraph of the page)

Discussion:

1. Please, revise the 3rd line of discussion section as: should read as “based on the self-reported change” …

2. Considering that the life skills and behaviors were self-reported by learners and instructors and could not be objectively assessed and/or compared to non-intervention setting, please revise the 1st paragraph of the discussion as: “We were able to highlight potential linkages between the SRHR information in the modules, the SRH knowledge acquired and the SRH motivation developed by the adolescents; the connection between these constructs and the life skills component of the curriculum; as well as the potential pathway for the SRH positive behaviors reported by the adolescents”.

3. Suggest using identified need in the 2nd paragraph of page 14 (With an estimated population of 10 million out-of-school adolescents, research and publications focusing on CSE for this population in Nigeria is limited) to promote the curriculum integration in non-formal education settings, but also discuss potential generalizability and relevance of the study findings beyond Lagos state in Nigeria.

4. Given the fact that there were clear areas of weaknesses in SRHR knowledge and gender-related attitudes in CSE-exposed settings, the manuscript could be significantly improved by suggesting actionable recommendations to further improve the CSE curriculum in respective areas.

5. As suggested under results section, please, analyze the potential effect of CSE on gender-norms based on sex-disaggregated study results.

6. The claim that “The Mixed-method design of this study gives additional validity to the findings based on the integration of the results from the quantitative and qualitative aspect” need to be strengthened with appropriate documentation of similar variables from different data sources. Please, add such results in the annexes to back up the claim or delate it.

7. Given that the learners were previously exposed to some of the topics during school education, it is important to note that the study did not examine/account for a previous exposure to CSE education among CSE-exposed and CES naïve study participants, as study limitation in the discussion section.

Conclusion:

1. Without the clear evidence and self-reported claims by study participants, the abstract conclusion should be revised as: “The study findings suggest that the integration of CSE into non-formal educational settings could improve SRHR-related knowledge among young people, promote inclusive gender norms and potentially contribute to improved SRH outcomes and related SDGs in Nigeria”.

Figure 1

1. Revise the tile of the figure as: Self-reported knowledge, skills and behaviors among instructors and CSE-exposed students (indicate number of participants in FGs and KIIs).

6. PLOS authors have the option to publish the peer review history of their article (what does this mean?). If published, this will include your full peer review and any attached files.

**Do you want your identity to be public for this peer review?** For information about this choice, including consent withdrawal, please see our Privacy Policy.

Reviewer #1: No

Reviewer #2: **Yes: **Tamar Chitashvili

---

## [Editor Report · Decision Letter 1]

27 Sep 2024

PGPH-D-24-00960R1

Comparison of comprehensive sexuality education knowledge and attitudes in CSE-exposed and CSE-naïve non-formal settings in Lagos State, Nigeria.

Dear Dr. sekoni,

Thank you for submitting your manuscript to PLOS Global Public Health. After careful consideration, we feel that it has merit but does not fully meet PLOS Global Public Health’s publication criteria as it currently stands. Therefore, we invite you to submit a revised version of the manuscript that addresses the points raised during the review process.

We look forward to receiving your revised manuscript.

Kind regards,

Tanmay Bagade, Ph.D., MS (O&G), MPH, MHM

Academic Editor

Journal Requirements:

Additional Editor Comments (if provided):

Please submit the document using tracked changes
---

## [Editor Report · Decision Letter 2]

7 Oct 2024

Comparison of comprehensive sexuality education knowledge and attitudes in CSE-exposed and CSE-naïve non-formal settings in Lagos State, Nigeria.

PGPH-D-24-00960R2

Dear Dr sekoni,

We are pleased to inform you that your manuscript 'Comparison of comprehensive sexuality education knowledge and attitudes in CSE-exposed and CSE-naïve non-formal settings in Lagos State, Nigeria.' has been provisionally accepted for publication in PLOS Global Public Health.

Best regards,

Tanmay Bagade, Ph.D., MS (O&G), MPH, MHM

Academic Editor